# Understanding healthcare providers' perspectives on barriers to accessing stroke care at a resource-limited hospital in East Africa: A qualitative study from Mnazi Mmoja Referral Hospital in Zanzibar

**Jutta M. Adelin Jørgensen**[1,2]*, **Elias Ditlevsen**[3], **Sanaa S. Said**[4,5], **Richard W. Walker**[6,7], **Dirk Lund Christensen**[1], **Karoline Kragelund Nielsen**[8]

1 Department of Public Health, University of Copenhagen, Copenhagen, Denmark, 2 Department of Training and Research, Mnazi Mmoja Referral Hospital, Zanzibar, Tanzania, 3 School of Public Health and Community Medicine, University of Gothenburg, Gothenburg, Sweden, 4 Department of Internal Medicine, Mnazi Mmoja Referral Hospital, Zanzibar, Tanzania, 5 Department of Internal Medicine, State University Zanzibar, Zanzibar, Tanzania, 6 Faculty of Medical Sciences, Newcastle University, Newcastle upon Tyne, United Kingdom, 7 Northumbria Healthcare NHS Foundation Trust, Newcastle upon Tyne, United Kingdom, 8 Health Promotion Research, Steno Diabetes Centre Copenhagen, Copenhagen, Denmark

* Kxz953@sund.ku.dk

## Abstract

Timely and appropriate care reduces the risk of mortality and disability after stroke. Despite high stroke incidence, little is known about the specific barriers to accessing stroke care in Zanzibar, East Africa. The aim of this study was to investigate healthcare providers' perspectives regarding the barriers to stroke care at the main referral hospital in Zanzibar. We used a phenomenological approach and conducted 14 individual semi-structured interviews with healthcare providers at Mnazi Mmoja Referral Hospital in Zanzibar. The interviews took place from April through September 2022. Thematic network analysis was applied to analyse and interpret the data. Three broad themes and eleven sub-themes were identified, relevant at specific stages in the patient's care pathway from deciding to seek care over receiving acute stroke care in hospital to accessing post-stroke care. These themes include health system barriers (medical supplies and equipment; staff shortages; attitudes and teamwork; organization of services; health education); patient-level factors (health literacy; relational factors; worries and feeling hopeless; financial constraints); and cultural context (stroke as a spiritual malady; trust, mistrust and power). Some of the sub-themes of barriers were similar to findings from other studies in both high and low-resource settings, such as shortages of medical supplies, equipment and staff, and sub-optimal organization of care. Other sub-themes were unique findings to low-resource settings, such as Zanzibar, and included relational factors and patients' perception of stroke as a spiritual malady. Interventions to improve stroke care should be informed by all these findings. Otherwise, focus only on removing barriers related to availability of stroke treatment in hospital may divert attention from significant cultural factors that affect health care seeking behaviour.

**Data availability statement:** Following the guidelines from Zanzibar Medical Research Ethics Committee the participants were guaranteed the highest level of confidentiality, and that collected data would not be shared outside the research team. Request to access data can be made through the Zanzibar Medical Research Ethics Committee under Zanzibar Health Research Institute (www.zahri.go.tz) directing an email to the chairperson on zahrec@zahri.go.tz.

**Funding:** The author(s) received no specific funding for this work.

**Competing interests:** The authors have declared that no competing interests exist.

## Introduction

It is estimated that every ten seconds, an adult in sub-Saharan Africa (SSA) develops a stroke [1]. While other regions over the past decades have experienced a decline in stroke incidence rates, SSA has experienced an increase, and today the world's highest stroke incidence rates are found here [2]. Several factors are proposed to contribute to high incidence and worse outcomes after stroke, including differences in cardiovascular risk factors, stroke prevention services, access to healthcare, social support, and socio-cultural attitudes towards stroke and disabilities [2–7].

The growing number of people with stroke places an increasing demand on health services in SSA to provide quality stroke care, but studies suggest that the uptake of evidence-based stroke care interventions in the region is lagging [8,9]. Studies from high-income countries have assessed barriers to implementing national stroke guidelines [10], treatment-specific guidelines [11,12] and new stroke treatments [13–15]. Only few studies have been concerned with perspectives from countries in SSA, including two studies from Ghana [16] and Malawi [17], respectively. These studies highlighted various challenges in stroke care that were unique to a low- or middle-income setting in SSA. As the stroke burden in SSA is still increasing [2], it is critical to identify barriers on both service uptake and delivery side that are unique to this setting [16] in order to inform development of contextually meaningful and effective interventions.

In Zanzibar, stroke incidence is high with an annual rate of 286.6 per 100.000 adults [18]. Overall mortality after stroke is high at 38.2% at 28-days raising to 59.0% at one-year post-stroke [19]. Routine data from hospitals suggest low uptake of specific evidence-based stroke care elements at hospitals including use of neuroimaging and physiotherapy [20].

Reasons for the low uptake have not yet been investigated. A pilot study was conducted in 2020 involving medical doctors at Mnazi Mmoja referral Hospital (MMH). The study suggested several barriers to accessing acute care being present both at patient level, healthcare worker level, hospital level, and related the wider society [21]. We are building on this work and aim to explore the various thematic areas and stages across the patients' care pathways in which obstacles to accessing care occur, as perceived by those providing the care.

In doing so, we apply the definition of healthcare access by Levesque which conceptualises access at the interface of health systems and patients as 'the opportunity to identify healthcare needs, to seek healthcare services, to reach, to obtain or use health care services, and to actually have the need for services fulfilled' [22]. This approach understands access by characteristics of the health system (approachability, acceptability, availability and accommodation, affordability, and appropriateness) and corresponding abilities of the population (to perceive, seek, reach, pay and engage with) which have to be present to turn opportunity into fulfilled healthcare.

In the current study, we are focussing on the perspectives of the health system actors.

The specific objective is therefore to investigate the perspectives of healthcare providers on barriers to accessing timely and appropriate stroke care across the care continuum at MMH in Zanzibar.

This study is performed as part of a larger project, ZanStroke, conducted from October 2019 through September 2022 in Zanzibar.

## Materials and methods

### Study design

Qualitative research is well suited for illuminating the various factors and complex dynamics that impact treatment and shape health outcomes and can serve as a starting point for

thinking about underlying structures and systems that go beyond quantifiable and mechanical factors [23]. This qualitative study applied a phenomenological approach and utilised semi-structured interviews to explore health care providers' perceptions. In the study, we focussed on healthcare providers' understandings and interpretations of their lived experiences with providing stroke care. We did not seek to assess the veracity of the health care providers' claims, assess their knowledge around stroke, explore their perspectives on what constitutes quality stroke care, or quantify care processes given [24,25]. Semi-structured interviews were chosen due to the sensitive nature of the topic, potential critique of the health system and its actors, and the need to capture diverse professional perspectives without hierarchical dominance. They allowed in-depth exploration of individual views, free from peer influence or fear of judgment, ensuring privacy and confidentiality. It was appropriate as the primary objective of the study was to understand the healthcare providers perspectives on barriers to stroke care. Through a partnership between a non-native researcher (JAJ) and a native researcher (SSS) we embodied a balanced approach to qualitative inquiry, drawing on the strengths of both insider and outsider perspectives. The non-native researcher's prolonged immersion provided familiarity with local norms while maintaining an external lens that facilitated access to potentially sensitive data. As noted in discussions on power dynamics in research [26], this outsider role can foster participant openness due to perceived neutrality. Conversely, the native researcher contributed essential cultural expertise and linguistic fluency, ensuring the research captures nuanced local realities. While not claiming to be ideal, this collaboration strengthened cultural validity and enhanced ethical engagement, acknowledging the critical need for equitable input from those rooted in the community in which the research took place.

We are aware of, and have attempted to 'bridle', our own pre-understandings [24] and to be particularly mindful about issues related to cross-cultural research by using a reflexive approach to the research process. This includes revisiting our interpretations, learning from our informants and other researchers, and being self-reflective on our own background and biases.

## Study setting

The study was conducted at MMH, which receives approximately 90% of admitted stroke patients on Unguja island, which has 1.3 million inhabitants [18]. In public health facilities, there are no user-fees, but resources are restricted and frequently patients must pay out-of-pocket for services or medical supplies from elsewhere. The 776-bed hospital is the main referral hospital in Zanzibar and has general medical wards and intensive care, but no specialised stroke units or stroke specialists. One special ward caters for neurosurgical patients, and the physiotherapy department employs several physiotherapists, two occupational therapists, but no speech therapists. Several social workers and counsellors work in the hospital buildings, some of which, like the general patients' wards, are very old and in need of repair, while other departments like radiology are newly constructed and air-conditioned. Wards hold up to 20 beds in each room separated into compartments of 4-6 beds, each with a bedside table but no mosquito nets. Most of the daily work on the wards is undertaken by intern doctors rotating between different departments, supervised by house doctors and specialists (when available), and by the few permanent ward nurses and hospital attendants. Many health science students are present at the hospital during daytime, and it is common to see 20–25 people following the ward round team from bed to bed. Family caregivers are always present at the hospital and waiting on the balconies and corridors when ward rounds are conducted, unless the patients cannot answer for themselves; then they stand or sit at the foot end of the bed, as there are only few if any chairs available on the ward. During the three daily visiting hours,

family, friends, neighbours, colleagues, and others arrive to visit the patients and to deliver food, drinking water, fresh bedsheets and other supplies. The wards and corridors become congested, and it is challenging to move around easily not to mention transfer patients in beds between the wards or do any bedside work. Patients with stroke were on average admitted to hospital for a few days and then discharged home to untrained family caregivers and without assistive products, and there was no systematic follow-up of stroke patients after discharge.

## Informants

The inclusion criteria for informants were being a healthcare provider employed at MMH, with any professional background and seniority, and who in the last two months had been frequently involved in providing direct care to stroke patients. We avoided narrowly defined selection criteria as they can negatively affect the ability to recruit informants [27] and the only exclusion criteria was being member of the ZanStroke research team. Purposive sampling was chosen to align with the study's objective of understanding healthcare providers' perspectives on barriers to stroke care at Mnazi Mmoja Referral Hospital. It ensured the inclusion of participants with direct, recent, and relevant experience, and guaranteeing maximum variation by capturing diverse perspectives across professional roles, seniority levels, and genders [26]. This approach, suited for phenomenological studies, focused on participants with lived experiences, ensuring findings were deeply rooted in real-world insights. It offered a structured, strategic alternative to random sampling, enhancing the study's depth, relevance, and rigor. Practically, purposive sampling was efficient for this low-budget study, avoiding irrelevant participants and unnecessary data collection, reducing transcription and analysis workload without compromising data quality.

Informants had no prior relationship with the interviewer (JAJ) but in her capacity as the principal investigator of ZanStroke, JAJ had frequently visited the hospital for clinical data collection prior to the current study, and some informants were familiar with her as a medical doctor and researcher.

The recruitment plan had two elements. First, during the interviewer's observations at the hospital, potential informants were identified, approached face-to-face, informed about what participation entailed, and given the chance to ask questions. Then, considering the possibility that not enough informants would be enrolled using this approach alone, recruitment through snowball sampling was enabled [28]. This meant that each informant interviewed was invited to recommend somebody they believed should be recruited and interviewed for more perspectives. New informants were recruited until code saturation was obtained [29,30].

A consent form was signed by both parties before the interview. One of the approached potential informants declined participation for unknown reasons. Table 1 shows informant characteristics.

## Interview guide

An interview guide in English was designed with primarily open-ended questions (S1 Text). The guide was thematically organised to explore different barriers related to uptake of services, delivery of services, interface between uptake and delivery side, and the broader context where healthcare took place. It covered several domains, including professionalism, organisation and management, human resources, and individual patient/family/health provider factors. These were based on review of existing literature around barriers to providing quality stroke care, which consisted of studies from both high- and low-resource settings. It was also informed by the pilot study [21] as well as multiple encounters with stroke patients and their families where they had shared their experiences. This led to specific open-ended questions

Table 1.  Informants characteristics, summarized.

| Education | Occupation |
|---|---|
| Medicine | Intern doctor |
| Medicine | House doctor |
| Medicine | Intern doctor |
| Medicine | House doctor |
| Medicine | Consultant |
| Nursing | Ward nurse |
| Nursing | Ward nurse |
| Nursing | Supervisor |
| Nursing | Ward nurse |
| Nursing | Clinic nurse |
| Nursing | Ward nurse |
| Physiotherapy | Physiotherapist |
| Psychology | Counsellor |
| Social work | Counsellor |
| **Age-group** | **Pax** |
| 20–30 | 9 |
| 30–40 | 3 |
| 40–50 | 2 |
| **Sex** | **Pax** |
| Female | 10 |
| Male | 4 |

around relational factors and power dynamics, socio-cultural beliefs and practices, and traditional/spiritual/herbal medicine.

The interview guide was tested on two healthcare workers and amended prior to data collection. This was done to enhance clarity of the questions, increase relevance and focus, and ensure getting well around the topic while staying within the desired time-limit. It involved removing or adding questions, rephrasing questions and probes, and reorganising the sequences of questions in the guide. Iterative amendment of the interview guide with adjustment of questions was undertaken as the study progressed and we gained new insights into the topic. The purpose was to uncover all important aspects around barriers to stroke care provision while ensuring fidelity with the original intention of the questions in the guide.

## Data collection

Interviews were conducted from April-September 2022 by JAJ, a medical doctor from Denmark who is fluent in Swahili, trained in qualitative research methods, and has extensive experience in health systems work in East Africa, including Zanzibar, where she has lived and worked for more than 10 years.

Interviews were conducted one-to-one. One interview had to be conducted by telephone due to the informant's busy schedule hence inability to commit time for a physical meeting, while the rest were conducted face-to-face. The informants chose the time, date, and location for the interviews. All interviews were audio-recorded after obtaining written consent from the informant. The one interview conducted by telephone, and the first half of the interview with a house doctor, were not audio recorded in absence of written consent for audio recording.

All interviews were conducted at the study site, typically in a nursing office or a consultant's office. These offices were adjacent to the patient wards and contained a desk, a cupboard, a sink, and a couple of chairs. The windows were open due to the heat, but the doors closed, and occasionally the interview got interrupted briefly by someone entering the room to give a message or pick up something. Two interviews took place outdoor in a shady, quiet place in the garden outside the hospital's main building. Field notes were taken before or after the interview and focussed on non-verbal things and impressions during the interview situation and observations from the wards and offices while waiting for the informant to arrive. Interviews were conducted in English, Swahili or a mix of both languages as preferred by the interviewee and lasted from 25 to 45 minutes, and there were no repeat interviews.

## Analysis

A thematic analysis approach was applied in the data analysis [31,32]. All audio-recorded interviews were transcribed verbatim by the interviewer (JAJ) and read through several times to ensure consistency with the spoken word. Transcripts of interviews done using Swahili were analysed in the original language and not translated into English. During the first readings, emerging ideas and themes were noted in the margin. A non-exhaustive list of potential codes and themes was then developed using English language, based on the study data and supplemented by JAJ's experience and existing literature in the field [11,16,17,33]. During further readings, meaning units were coded and organised in thematic categories and re-organised in overall themes. All coding was from this point onwards done using English language and applied to the verbatim transcripts of interviews whether English, Swahili, or a mix of the two languages had been used during the interview. The analysis process was iterative and occurred concurrently with data collection, resulting in modifications and adjustments to the interview guide as previously described. New codes and themes emerged from incoming data, leading to changes in the coding and organization of previously collected data. Along with this there was a continued reflexive dialogue with the Zanzibari co-investigator (SSS) on the data material and personal reflections throughout the study. When unsure of how specific meaning units could be understood, or if there were alternative understandings to the text that I had not been sensitive to, these were also discussed with SSS as well as with another Zanzibari medical doctor with interest in stroke research.

An example of meaning units, codes, sub-themes, and themes is shown in supplementary material (S1 Table). Code generation and themes were in detail discussed with last author KKN, a public health researcher from Denmark with extensive experience in qualitative research in both high- and low-resource settings. KNN also read transcripts from the interviews conducted in English.

After analysis was completed, quotes by non-English speakers were translated into English by JAJ while drafting the manuscript.

Nvivo 12.0 (QRS International, USA) software was used to assist the data analysis process, and the reporting of findings follow the overall guidance in the consolidated criteria for reporting qualitative research (COREQ).

## Ethics

The study was approved by the Zanzibar Health Research Institute under the ethical clearance for ZanStroke (ref ZAHREC/02/29 July/42). All informants were assured that any questions regarding the study were welcomed at any time, and that they were free to not answer questions, or to discontinue participation in the study should they so wish, even after their interview, and that this would have no implications.

## Findings

Based on the healthcare providers' perspectives on barriers to timely and adequate care, three overarching themes and eleven sub-themes were constructed along the different steps of the stroke patients' care journey of identifying a need and deciding to seek treatment in hospital, receiving acute stroke care once in hospital, and accessing post-stroke care. The three overarching themes were (1) Health system barriers; (2) Patient-level factors; and (3) Cultural context.

There was an overlap in themes between the different stages in the stroke patient care journey, with different sub-themes dominating at each stage (Table 2). The findings are organized in the below sections so that each theme will be presented together with its sub-themes.

## Health system barriers

The first theme is concerned with the various types of resources informants identified as needed to provide care, and systems that must be in place in the hospital for these resources to be translated into active care for patients. Five sub-themes appear under this theme, specifically: Availability of medical supplies and equipment; staff shortages; healthcare providers' attitudes and teamwork; organization of services; and health education.

**Availability of medical supplies and equipment.** The insufficiency of working equipment and medical supplies in hospital, ranging from medication, cannulas and catheters to more trivial work essentials like plasters and gloves, was consistently identified by all informants as one of the main barriers to good care.

Shortages of equipment like blood pressure monitors and oxygen concentrators, and malfunction of CT-scanners and laboratory equipment, temporary in nature but of a recurrence and duration that resembled chronicity, were also consistently mentioned. One nurse went on to describe how the entire ward of around 30 patients currently only had one functioning blood pressure machine so, even disregarding other limitations, measuring blood pressure every 15 minutes as per protocol for the critically ill stroke patients was practically impossible. Some informants elaborated on how the unavailability of medicines, medical supplies and functional equipment constituted an immediate barrier for the patient to receive care, but also affected the health care provider's ability to perform their jobs satisfactorily.

> *Reagents to perform a full blood count? It is just out of stock [laughs] it has been out of stock for a year now. You know, we just survive here (house doctor#3)*

**Table 2. Themes and sub-themes of barriers to quality care as they emerged along the stroke patients' care journey.**

| Themes: | Health system level | | | | | Patient-level factors | | | | Cultural context | |
|---|---|---|---|---|---|---|---|---|---|---|---|
| Sub-themes: | Availability of medical supplies and equipment | Staff shortages | Attitudes and teamwork | Organization of system and services | Health education | Stroke awareness and health literacy | Relational factors | Worries and feeling hopeless | Financial constraints | Stroke as a spiritual malady | Trust, mistrust and power |
| Deciding to seek care | | | X | | X | X | | | | X | |
| Receiving acute stroke care | X | X | X | X | | X | X | X | X | | X |
| Accessing post-stroke care | | | X | X | X | X | X | X | X | X | |

*Patients' care journey*

> *You can only observe the patient. The patient say I don't have money. Okay, I can help you to do the full blood count, I say it costs less than 10.000 shillings (intern doctor#4)*

Many informants described how patients, family caregivers, and health care providers together were improvising to acquire the needed medical supplies and diagnostic tests. This could be done either by borrowing supplies from other patients, from other departments at the hospital, or procuring them at private pharmacies, and blood specimens were sent to private laboratories for analysis.

Informants described how health care providers often were involved in coordinating these 'fixes' and occasionally paid out of their own pocket to compensate for the shortages and enable them to do their work. However, by far most of the time it was the family caregivers who were responsible to financially and logistically provide this, and the healthcare providers were depending on families to fixing this so that they could provide care for the patients.

> *The relatives are waiting in the garden. If you need them, you call them (…) because here at the hospital not everything is available. Many things are not available, so the relatives must bring them (#7)*

> *At our hospital there is every day something that is missing [laughs slightly]. Like medical supplies, medicine, you miss it. So sometimes we want to get it through the relative. If there isn't good cooperation with them, you can't provide care, because you cannot spend your own money on it. You must provide proper care, but it is difficult (nurse#5)*

One nurse described the additional challenges during nighttime when private pharmacies and laboratories were closed. Even when the treatment was considered essential by the family, and they had the funds to buy medicine and other supplies and tests from private providers, they could not access it until the morning hours. This could lead to delays in diagnosing, starting medical treatment, or providing care, which again could lead to more complications to manage and worse outcomes for the patient.

**Staff shortages.** The first thing that many of the informants mentioned as a barrier for patients to access care was the absolute shortage of healthcare providers in hospitals. The shortage included primarily frontline nurses and doctors, leading to a high patient-to-provider ratio, but also shortage of orderlies, porters and radiographers and the like constituted a bottleneck to providing care. One medical doctor described how shortage of orderlies meant that blood samples were not delivered to the laboratory on time, and diagnostic results would not automatically be returned to the nurses and doctors on the ward. Another informant described how the shortage of porters meant that either doctors or nurses had to spend time transporting patients from one department to for instance the radiology department and back again having less time to focus on their core job tasks, or alternatively that the investigations would not be done. One doctor described the impact of shortage of radiographers by saying:

> *Only one radiographer is on duty during night shifts, so if you go to the CT scanner you might not find him there. He has many X-rays to take which is more of an emergency than the CT scan. If you are very lucky he is at the scanner and you will get the emergency CT scan done, but if he is not there when you bring the patient, you'll have to continue to the ward and once they are there, it is very difficult to get a CT scan organized.. (intern doctor #1)*

According to most informants, the high patient-to-provider ratio affected the healthcare providers time with the patients and ability to provide the required care, including monitoring

progress and providing health education, and it caused delays in attending to patients' needs –
both the more basic bodily needs as well as the specific medical needs.

> *Sometimes the family caregiver is not present to help with feeding or assist with toilet vis-
> its. If you are only two staff, you cannot provide care at those times. Often you finish your
> shift and you feel like wow, I only finished half of my job.. (nurse #7)*

> *On our ward, there's many patients and only one doctor. Then, when new admissions
> are coming in, some of them quite serious, the ward gets full, and the doctor will see those
> patients and it is not possible [for the doctor] to do any close monitoring of the [stroke]
> patients (nurse#9)*

Only two informants identified how gaps in general knowledge and skills around stroke
among healthcare providers currently providing stroke care posed a challenge to care, either
indirectly, or directly as shown in the quote below. However, several informants identified the
unavailability at the hospital of health professionals with specific stroke competences, such a
stroke nurses or neurologists, as impacting the ability to ensure quality stroke care and good
patient outcomes. With stroke specialists on the team, these informants described how more
competent and qualified treatment decisions could be made, processes could be organized
better and care would be of a higher quality.

> *You need a special physician, a specialist. When there's a cardiac patient, the cardiologist
> comes to do the ward round. When kidneys, the nephrologist comes to do the ward round.
> Stroke patients also need the specialist who knows what to do (nurse#6).*

**Attitudes and teamwork.** Working in an environment with chronic shortages of staff,
medical supplies and equipment were described by many informants as causing exhaustion,
demotivation, and fatigue, which affected the way they worked with each other as well as
how they worked with the stroke patients and their families. Some informants, in particular
nurses, described interaction with their immediate colleagues as 'perfect' and gave examples of
how they were going out of the way to help each other, of staying on after-hours to finish up
and not leave the next on shift with left-over work, and always received help and supervision
when asking for it. Other informants pointed at lack of good communication, a low degree
of collaboration between departments, and a low degree of team spirit as barriers for patients
to receive the care needed. Sometimes this poor collaboration would lead to conflict between
health providers or departments on whom to be responsible for the patient.

> *Teamwork is very very very important (…) but nowadays we argue because people are
> escaping the patient. They say 'Oh, this is your patient, not mine'. You ask for transfer to
> another department but their consultant doctor says 'No, I'll just come for a consultation
> but the patient will remain at your department' (nurse#6).*

Some informants described how they were already exhausted and feeling fatigued when
entering their shift, knowing that they had not the resources, system, support, or staff to com-
plete their job in a manner that was satisfactory to themselves nor the patients. One doctor
described how she coped with never-ending demands on night shifts where she would be
responsible for several wards together with an intern doctor while the senior doctor would be
on call from home. As a young doctor she would have entered the patient wards at night shifts
and be swallowed up trying to fulfil all needs and demands from family caregivers and patients
and would leave in the morning hours feeling 'like a zombie'. Therefore, she now would not

enter the patient wards voluntarily at night any longer and, if for some reason she had to go there, she would not respond to the family caregivers' requests but tell them that she was busy and would come back if she got time. She regretted how this affected the patients and their caregivers but explained that when demands were surpassing her capacity and ability to cope with them, and there was no support from her seniors, placing these boundaries was the only way she would be able to go to work.

An experienced ward nurse described how she coped when feeling overwhelmed by work and requests and could not leave the patients and the ward to get a pause. She would tell family caregivers who sought her attention that she could not do anything for them right now, that she had nothing to give to them, and that they would have to come back the next day.

> We have to serve the patients though I am left with nothing. I have nothing at all left, I cannot do anything. It's exhausting (senior nurse#2)

Other informants underlined how it was a personal decision *how* to cope with working in a chronically under-resourced environment, and that in the end it came down to which attitude the individual had to their job as a healthcare provider. A senior nurse described how they had experienced that 'setting your mind' on doing the level best under in the given circumstances would lead to better care and patient outcomes.

> Sometimes it depends on the staff, how hard you work to care for these patients. Because what I know is that if the staff in their minds are serious about a stroke patient they work hard. If they're lazy, the patient will die (…) It is a mindset; if they say 'I don't want any-one to die on my shift today' they will not have any deaths. (nurse#6).

Some medical doctors identified flaws in teamwork particular with nurses and auxiliary staff. Specific issues spanned from nurses forgetting to perform tasks to direct unwillingness to do what they were requested to. Parts of this were identified as no awareness of written job descriptions, while parts were consequences of being stretched way beyond capacity. This led to conflicts and overburdened those who felt responsible to make the system 'work' by placing additional workload on them, consequently impacting how they ended up providing care to patients.

> From the beginning it is the nurses' responsibility to care for the patient. Provide care such as bed baths, that's a nursing responsibility. [Is that also part of your job here?] Yes, it is part of my job. But nowadays it is not being done because there are so many patients and few staff; on a shift there is only one nurse to care for 30 patients (nurse#5)

> None of us have job descriptions (…) If on admission for instance the nurses take an ECG it will save you a lot of time in order to diagnose the patient. But if you see the patient and the ECG is not done you first have to go and make the machine work, take the ECG, interpret it; sometimes it is chaotic. It is like you have to do the nurse's job, the hospital attendant's job, and your own job, so at some point you are either frustrated or tired, and your performance is not good (house doctor#3).

Some senior nurses and doctors found that their colleagues had an overly pessimistic per-ception of the potential of recovery after stroke, and that this in turn unintentionally led to lower willingness to provide care as the patient was already 'given up on'. Poor care, poor outcomes and poor perception on recovery potential were hence synergistically affecting each other.

**Organization of system and services.** Systemic problems also materialised as a lack of effective coordination between different levels of the health care system. Several informants described how the absence of an effective referral system led to MMH receiving patients who could have been treated at lower levels of healthcare. As an example, one intern doctor described how he had received patients with uncomplicated urinary tract infections or anaemia who in his view could be managed at a lower-level hospital. Consequently, the medical wards at MMH became congested and staff attention to the most critically ill patients diluted.

The same informant described how delays occurred when a lower-level health facility did not promptly refer critically ill stroke patients but waited days before transferring the patient to MMH, either due to ineffective referral practices, lack of skills to identify stroke, or a combination of both as in the example below:

> *[They] come from nearby hospitals. And most of the time they come late, I can say that. They come maybe after 3 to 4 days, after presenting with neurological deficits, being unconscious or convulsions at home and then they visit the nearby hospital and are misdiagnosed (intern doctor #4)*

The absence of a well-functioning referral system also challenged down-referral of stroke patients who were discharged from hospital after end of treatment and needed follow-up at lower levels of care. Upon discharge from hospital, patients were provided with a one-page discharge note advising them to visit a hospital outpatient clinic, such as hypertension clinic or physiotherapy/rehabilitation clinic, after a specific number of weeks. As there was no system in place to follow up if this advice had been understood and followed, the perception of many nurses and doctors was that people frequently delayed or never showed up in clinics. This lack of continuum of care was cited by many informants as a main barrier to both rehabilitative stroke care and prevention of new strokes:

> *Actually, after patients leave the ward, we have no follow up. We just tell them to come to the physiotherapy on Wednesday and Mondays, so there is no follow up. Actually, some of them are not coming back. There is no follow up at all. Maybe the patient then is re-admitted with a new stroke (intern doctor #4)*

> *One of the big problems is that the patients and their relatives are not informed about the [stroke] diagnosis and what to expect, the prognosis. They do not know they should come for rehabilitation (house doctor#11)*

One senior staff pointed to the organization of care in-hospital with no dedicated stroke unit and inconsistent adherence to guidelines as an obstacle to ensuring all stroke patients received standardized care. Within a stroke unit, multiple disciplines can work together to tailor care to the individual patient, ensuring best practices are implemented and standard protocols followed. Instead, they described how care at MMH tended to be 'individualized' meaning it depended on who the health care provider on duty was and how much effort they put into the care, not on the needs of the patient:

> *Sometimes I see that they have done it all according to the book; all investigations have been done and all treatments given. But others don't follow through, they don't check up and they only do half. It depends on the individual doctor, how well they perform their job (consultant doctor #14)*

**Health education.** Many informants perceived health promotion and providing health education as crucial aspects of a healthcare provider's role, not only for patients and their families but also for the broader community. Yet, they agreed that health education was often neglected, and that the general population received inadequate health education, particularly regarding stroke. This lack of education resulted in a widespread perception that effective stroke treatments were unavailable, the potential for recovery was minimal, families had already given up on patients, and that notion of powerlessness could affect both service uptake and delivery

> *They think that in hospital they won't do anything, they think they just wait for the patient to die in the hospital because they don't have any health education. They think there will not be done anything (nurse#6)*

Three of the informants, all younger doctors, described how they always made sure to explain to patients and their family caregivers what stroke is, what it is caused by, and how to treat and prevent a new stroke in the future. One of them explained how this to him was a prerequisite for patients being able to adhere to treatment, and another described how she, when unsure if the nurses would have time to explain and remember to dispense the medicine, depended on the patient and their family caregiver to know what medication to use and when to administer it. In that sense, health education became not only about knowing, but also about enabling 'getting it done'.

One nurse working with rehabilitation compared health education around stroke to the health education delivered through the antenatal care program, which also took place at the hospital's outpatient department. Here, the waiting area was large, and it was planned that each morning started with group health education around topics related to pregnancy, childbirth, and young children, before the individual services were delivered. She perceived this model of health education as very successful as the expecting mothers were taught and would follow what they were learned. The situation for health education around stroke was according to her quite different, starting during admission to the medical ward:

> *Often the time for delivering health education is planned at the time of discharge, you see? They are going home, you meet them and you have so much information to give them on what to do next. Therefore, I would say that health education is very behind. And we struggle to do this work. If you meet a staff they is tired, alone on the ward caring for all those patients. They cannot also provide health education. (…) The rooms at the outpatient clinics are not conductive, the space is too small, and people are waiting in the corridors. With the right space and right planning, we could do health education, and people would leave the hospital with this knowledge and even pass it on in their communities. Its all about planning. Now the knowledge stays with the staff.. (nurse#8)*

Another nurse described how in her perspective, people would adhere to the prescribed treatment had they been educated around it; but now the lack of health education around stroke impacted post-stroke care and prevention of new strokes:

> *Many people have not gotten any health education. They use [blood pressure lowering] medicine for a while, feels better, and then stop using it. It is because they did not get health education. If they get education, they will change behaviour*
>
> *(nurse #7)*

## Patient-level factors

The second theme is concerned with factors that are closely related to the individual patient and their family, with or without an underlying systemic root cause. Four sub-themes emerged under this theme: stroke awareness and health literacy; relational factors; financial constraints; and worries and hopelessness.

**Stroke awareness and health literacy.** Where the subtheme above, health education, was concerned with the health education being delivered by health system actors, this sub-theme is concerned with the extent to which patients and their families understood the signs and symptoms of stroke, their knowledge of appropriate actions to take during stroke events, and their ability to search for, critically appraise, and apply relevant information around stroke to their own situation.

Informants from various professions perceived patients and their families as generally having low levels of awareness of stroke being a biological disease. They found that many patients lacked basic understanding about the nature of stroke, its risk factors, warning signs, and the urgency of seeking prompt medical attention. According to informants this, together with a limited knowledge of the treatment available for stroke in hospital, hindered timely recognition and response to stroke symptoms by patients and their families, leading to delays in seeking and accessing critical care.

> *You know, the community did not receive education on this illness [stroke], they say here in Zanzibar it is a spiritual possession. If a person gets this, they keep him at home until he develops a disability (nurse#8)*

> *They think they will just wait for the patient to die in hospital, because they don't have any health knowledge. They think there is nothing to do (senior nurse #6)*

Aside from a lack of knowledge, several informants perceived many people to lack the ability to critically reflect both on health messages that reached them from various sources, and the credibility of the senders of these messages. One physiotherapist described how some people with stroke used traditional practices like massages or herbal treatments at home and declared it led to improvement. According to him, when he assessed them, he discovered that they had not improved at all since the therapy had not resulted in the brain being trained to use new neural pathways, but this was not comprehended by the patients, who uncritically accepted what they had been told by their families and friends at home.

> *You know, people don't understand how these things works. They are being told other things at home which they believe. There is a lack of understanding..*

> *(physiotherapist #13)*

Several informants observed that low health literacy was both associated with delayed recognition of stroke symptoms and misconceptions around stroke, and with suboptimal adherence to medical advice among stroke patients both while in hospital and once discharged home. According to them, patients with limited health literacy struggled to understand medication instructions, and this lack of comprehension led to improper self-care practices and inconsistent medication use as a nurse exemplified in the quotes below.

> *The follow up in the community [after discharge from hospital] is not really working. Due to the patients themselves. They don't follow our advice. We tell them once you have been diagnosed with hypertension the treatment is ongoing, it is for life, do not stop your medication but they usually do that anyways (intern doctor #1)*

*If you give them health education, if you educate them, then they will leave behind the harmful beliefs they have [about treatment] and follow your advices. If they get education, they will change behaviour. But there are also others who still cling on to their harmful beliefs (nurse #9)*

**Relational factors.**  This sub-theme is concerned with the relationship between stroke patients and their families. The relationship may include social conflicts, disagreements around care decisions, competing priorities in the family, and similar relational issues which can constitute barriers to adequate and timely care.

Many informants either indirectly or directly described how healthcare was not an individual or private matter involving only the person who had fallen sick, but rather a social process involving the wider family and in particular, the family head. Family dynamics varied, and while many patients had good support from their families, other families were in the informants' views dysfunctional with some patients experiencing neglect and abuse. Several informants identified how existing conflicts between the patient and family members could constitute a significant obstacle: if no family member was willing to escort the patient to the hospital, stay to care for them while admitted, or provide care at home after discharge from hospital, this patient would not receive the needed care.

*Often the obstacles to good care are due to social problems, to family problems. If there is a conflict between the relatives it is hard, it is so difficult to care for that patient. Because we lack cooperation from them.. (nurse #5)*

*While admitted to the ward he told me that he had no one at home who would care for him (...) We could see that. His relatives were with him at hospital, but they did not care properly for him (counsellor #12)*

Most informants also expressed an emphasis on *relatedness* in decision making around stroke care. Rather than the person with stroke being the decision maker, decision making was a consultative process including family members and potentially also healthcare providers and trusted family friends. One informant explained that in ideal situations a patient and their family's views would be aligned when it came to deciding on hospitalization, treatment, and rehabilitation; but in case of disagreements the decision power laid with the head of the family. Sometimes the extended family got involved to mitigate and see if the wish of the person with stroke to be admitted to hospital could be accommodated, and some informants had examples of themselves mediating in similar situations in their own communities and described how they, in lieu of their health education, would be listened to and their advices followed, at least when they were closely related to the person with stroke. When asked about what would happen if the stroke patient wanted to go to hospital but the relatives did not approve, one informant said:

*Then he will have to remain at home. But in some families perhaps another relative is approached to see if they will accompany him to hospital (nurse#6)*

The support from the family was widely noted by informants not only to get to hospital and access care, but as a general foundation for practical and emotional wellbeing. One informant described how disagreements between a patient and their family also could raise a dilemma for healthcare providers in situations where the patient preferred to remain admitted and receive care in hospital, but the family requested the patient to be discharged home. The informant explained this dilemma as paying respect to the patient's autonomy

in decision making could cause a conflict between the patient and their family on whom the patient was depending for care once back home. Practically, this conflict could undermine the willingness of the family to provide good long-term care, so some informants experienced a dilemma both between prioritizing the patient's need for autonomy or relatedness, as well as between prioritizing the care benefits to the patient short-term while in hospital, or long-term when at home.

> *The dilemma is if the patient accepts that he got a stroke and needs to remain for treatment in hospital, but his close relatives do not share this belief and want him discharged home (...) the relatives have the final say. But we first talk to a social counsellor (nurse#2)*

A central element to all care at hospital was that each patient needed to be accompanied by a caregiver when admitted to hospital as there were no trained nurse assistants at the hospital. Hospital orderlies were helping in areas of monitoring patients, cleaning premises, fetching medicine, and delivering samples to the laboratory, while a family caregiver would be the person responsible for assisting the patient with issues related to personal hygiene, feeding, and mobilizing.

Also, family caregivers were responsible for bringing food and drinking water; bringing in fresh bed linen; procuring medical supplies when the hospital was out of stock; organize for diagnostic services outside the hospital when needed, and a range of other tasks that would otherwise fall under a nurse assistant's job tasks.

Nurses and doctors perceived how competing priorities in the family, such as income-generating activities or how to spend time and family resources, could become barriers to care, as it led family caregivers to request the patient being discharged home from hospital before finalisation of diagnostics and/or treatment.

> *They say we can take care of him at home, yes they say, because we are busy, we go to work, someone has to sleep here, if we go home we will take care of him. But from seeing them on the ward you see that they don't really give that care so if they go home do you really think they are going to give attention to the patient? (intern doctor #1)*

Likewise, a physiotherapist noted that especially people who had severe disabilities after stroke could not access rehabilitation services if their family members were not available to escort them back and forth to the physiotherapy, or help with exercises at home, as there were no community- or homecare services. Occasionally families could compensate for lack of time by asking the physiotherapist to come and do exercises in the patient's home and he underlined how this was a relational thing between individuals:

> *You know, any home visit is between the relatives and the physiotherapist. For humanity. It doesn't happen every day but one day a week; it is on the work roster, during working hours. I often go on Saturday or Sunday. The car and the fuel are paid by the relative, not by the hospital (physiotherapist #13)*

**Worries and feeling hopeless.** As previously described, it was the informants' experience that getting a stroke could be regarded as a situation surrounded by little hope of survival and recovery. One doctor described how she would counsel a patient and their family, explain that the disability would be permanent, and noted how this made them feel sad and hopeless. In her view, a lack of hope could affect whether the patient would remain in hospital to receive care or would 'give up' and return home.

*They already lost hope. Because we told them that this condition cannot be reversed, the patient cannot stand up like he used to do before. So perhaps because of the ignorance of the society they think 'we are just here in hospital wasting our time. If nothing can be done anyways, then we can just as well go home and stay with our patient, treat with massages and oils; maybe THAT will help..' (intern doctor#1)*

Some informants identified the sudden separation of body and self that a person with stroke experienced placed them in a situation of psychological distress and worries that they would become dependent, would be neglected, or would not be able to provide for the family. This could lead to a regular depression and lack of initiative to engage in treatment and rehabilitation activities, or to denial of the diagnosis, causing low adherence to the prescribed treatment.

*In general, the challenge is that the stroke patient perhaps thinks his life has ended. That can affect psychologically. He might think about his family 'I cannot walk any longer, who will take care of me, people will distance themselves from me (...) It affects psychologically but also physically. First of all, it leads to a deterioration in recovery because his mind is preoccupied with a lot of things. You see, you are very slow because you don't feel free in your body with all those worries (counsellor#10)*

**Financial constraints.** This sub-theme is related to direct healthcare expenses as well as indirect expenses related to treatment.

As described, the public health system lacked resources and frequently all the necessary elements required for providing free services in public health facilities were not available. The barrier that a lack of money made to accessing care was consistently brought up by most informants as it practically was the patients and their families' responsibility to fill the gaps between what the patient needed and what the hospital could provide. At times, this led to the patients not receiving the needed care at all. At other times it impacted the ability to receive timely and adequate care when being admitted to hospital, and this sometimes had detrimental consequences. One intern doctor described a care situation, which he felt was not going well and concluded:

*It was time to insert a nasogastric tube but unfortunately the relatives were not able to buy it the first day, the second day, and then the third day then the patient died. The cause of death was maybe hypoglycaemia, the patient was not receiving any nutrition, only i.v. fluids. (Intern doctor #4)*

Challenges to pay for transportation to the hospital were frequently mentioned to pose a barrier to accessing rehabilitation services and follow-up care after discharge from hospital, but less so for accessing acute stroke care. However, nurses and counsellors perceived that financial constraints impacted how long the patient remained in hospital for care. Despite no daily bed-fee to pay while admitted to hospital, the daily transportation cost for the family caregivers travelling from their home to the hospital to bring supplies, as well as the cost incurred by caregivers when they exchanged shifts with one returning home while the other travelled to the hospital, was significant. If not covered, the patient could not remain in hospital but had to be discharged home before time.

*The problem is money. It contributes [to poor care]. To come to hospital and return home – some families cannot afford this, okay. Some have to come in the morning, others in the afternoons, some stay far away from the hospital. (socialworker#12)*

## Cultural context

This theme is concerned with how people in Zanzibar perceive and think about stroke in a general sense—the shared beliefs, ideas and attitudes towards stroke that exist in the society - and how they understand and respond to the condition when it occurs.

Two sub-themes were identified under this theme, namely, stroke as a supernatural condition; and trust, mistrust and power dynamics.

**Stroke as a supernatural condition.** Awareness, beliefs, and preferences for traditional therapies were identified by all informants as major barriers to successful acute stroke therapy and rehabilitation. According to informants, the common socio-cultural belief was that stroke, a condition which has no warning signs or gradual progress like for instance an infection has, was associated with supernatural forces or witchcraft. While symptoms were physical, the cause was thought to be spiritual, and as a result, treatment and cure was sought in that domain.

> *Because of the scenario, that he collapsed when he was at the bathroom, they were convinced that this was NOT a hospital-related illness. He did not know that he had hypertension, and he was not sick before collapsing; it happened all of a sudden, so it clicked directly to them that this could not be a hospital-related illness*
>
> *(intern doctor #1)*

Informants identified that some people with stroke would not seek care at health facilities at all and instead underwent various 'traditional' treatments at home. These treatments included, but were not limited to herbal medication, traditional Islamic medicine and prayers, massage with oils, or puncture of peripheral blood vessels to remove 'dirty blood'. Other people with stroke used these practices before, during or after hospitalization, which in one way or another was identified as having an impact on care by delaying start of, interfering with, or leading to discontinuation of hospital treatment.

> *Those who stay at home with the one who suffered a stroke say that the condition is due to something supernatural, a spirit, therefore they will first treat him at home and only when his condition deteriorates [high pitch voice] they will bring him to hospital (senior nurse#2)*

Informants portrayed themselves as people who extended professional knowledge into the social sphere and educated the surrounding community on correct actions to take when a stroke occurred, and people who, due to being educated, were listened to. At the same time, health providers were culturally embedded. Both the professional, biological perception - that stroke is caused by an interruption of blood flow to the brain which leads to neurological deficits and treatment should promptly be initiated at hospital, and social discourses in which stroke was perceived as a supernatural or spiritual illness, could at times impact how some health care providers would care for stroke patients, such as below:

> *Some doctors or nurses or orderlies tell the patients 'According to what I see this patient should not be in hospital. This is [a problem that is concerned with] our spiritual world'. Something like that. Yes, but they are only few. Because those who do that, in my view they do not have enough education about stroke or other hospital issues (senior nurse#6)*

**Trust, mistrust and power.** Diverging beliefs on stroke causes and treatment could escalate into conflicts between family caregivers leading to poor cooperation around the

patient, both on decisions on when and where to seek care, adherence to treatment, and to request discharge from hospital to pursue alternative treatment options.

> There is one patient, the relative want to take discharge on request. To go home and do spiritual or other treatment. The condition of the patient was not good. But they insisted. Seven relatives insisted 'We need to take our patient home'. We spoke to them but it was bad luck. (intern doctor #4)

Sub-optimal cooperation between health care providers and patients and their families was another frequently identified challenge. According to informants, both biomedical and traditional treatments were frequently used together. Families might have adjourned medical care for traditional treatments, or made attempts to combine both while hospitalised, which sometimes led to conflicts like the situation this experienced nurse elaborated on:

> It happened once at male medical ward; the stroke patient had been given some traditional medicine to drink. After finishing my evening shift, I waited, and they waited; in the end I chased away the traditional healer. But you see; then I was chased by the relatives who immediately requested the patient to be discharged home, and they left the hospital with him (senior nurse#5)

Mistrust and deception between families and healthcare providers was not uncommon. Where family members mistrusted the intention and ability of healthcare providers to treat stroke, healthcare providers mistrusted the family caregiver's ability to understand and make good choices on behalf of the stroke patient and to adhere to the prescribed treatment.

> [Poor cooperation is] if you give the patient medication and he is not swallowing it - he can even pretend and afterwards spit the tablets out and hide them under the mattress, and later you find that the blood pressure is still very high (senior nurse#2)

> The patient comes to the hospital and the relatives keep asking 'so what is the plan for this patient'. And we are just monitoring vital signs [because of shortage of medical supplies] (…) They think we are bogus. [laughs timidly] We have done nothing to help the patient. They get angry. 'The patient is just sleeping here. You have not given any medication… (intern doctor#4)

> Because sometimes you know the relatives can be the one who hide what happens at home. For example, when we see this patient with stroke we know if they have been doing these traditional treatments or not. By observing them. [How can you tell?] They come with so much oil applied to the skin, and they are smelling like they have used some traditional treatments, so we know they are doing this.

> (physiotherapist #13)

As already elaborated above, family caregivers had according to informants the decision-making power over when, where, and how the patient should undergo treatment. However, they also held a more direct power over health care providers and impacted their ability to care. The perspective of many informants was that without the family caregiver's assistance to buy medical supplies, send blood samples to laboratories outside the hospital, and multiple other small micro-managing actions, the health care providers were in a poor position to fulfil their professional roles of administering medicine, interpreting blood results, adjusting care plans, and educating about the condition.

## Discussion

### Summary of findings

This study's findings illuminate the multifaceted barriers that impact access to stroke care from the perspective of those responsible for providing it. The study showed that healthcare providers experienced several barriers to care related to the health system, patient-level factors, and the cultural context. Different sub-themes of barriers dominate at different steps in the patients' care journey from identifying a need and deciding to seek care in hospital, over receiving care, to accessing post-stroke rehabilitation.

The cultural context unveiled a complex interplay of different explanatory models for stroke, including supernatural beliefs, and a lack of trust between healthcare providers and family caregivers. This dynamic potentially influenced treatment-seeking behaviours and healthcare interactions. Delays and substandard care in hospitals were attributed to limited resources, overburdened and sometimes burned-out staff lacking specialized stroke training, and a lack of coordination in stroke services. These factors hindered timely and effective interventions when patients were hospitalized.

Furthermore, patient-related challenges, such as low health literacy among patients and their families, strained family relationships, financial constraints, and a perception of stroke as a hopeless, incurable condition, exacerbated difficulties in accessing and receiving appropriate acute stroke care and rehabilitation services.

### Comparison with previous literature

One of the most prominent findings in the study was the health system's structure and resources gaps with stockout of medical supplies, malfunction or lack of equipment, and shortage of trained staff. These factors had a large and tangible impact on healthcare providers' ability to perform their jobs and care for patients. This has also been identified in a variety of other settings, also in high income countries [11,16,17,33]. In this regard, the largest differences between studies from high and low resource settings, including our study, are the absolute degrees of resource constraints, and to what extent they impact care.

In our study, personal hygiene, feeding, and similar tasks were seen by healthcare providers as largely domestic activities and were almost exclusively performed by family caregivers. This was possibly a way to provide a temporary 'fix' to the shortage of nurses as these tasks can be undertaken by lay people, freeing time for healthcare providers to focus on other tasks. Small improvisations and 'fixes' to compensate for all sorts of shortages were constantly taking place at MMH, mainly performed by family caregivers but also engaging healthcare providers.

A study from Sierra Leone on patient care pathways found that coordination of care processes in the severely under resourced hospital setting could be understood as a form of repair work. Healthcare providers and patients' families had to improvise this repair work to compensate for the lack of organized system maintenance in order to make the system work well enough to enable patients to receive care. This 'individualization of responsibility for making the system work' [34] placed an extra burden on healthcare providers and families, and it was vulnerable to interruptions or discontinuation. This seemed also to be the case in our study where the individual healthcare provider and patient's family were responsible for the needed medical supplies, cash or logistics being in place so that the patient could be cared for. Understandinghow these responsibilities for repair are distributed is important so that more sustainable and fair systems for repair can be developed and barriers for access to treatment can be minimized..

Yet, it is important to remember that care is also a cultural phenomenon [35,36]. In Zanzibar it is generally considered inappropriate to let an 'outsider' care for a (naked) elderly

relative as it is the duty of an adult child to care for their parent. Other cultural reasons for nurses not to perform tasks related to hygiene and feeding has been found in other settings [36,37] but more research would be needed to draw any conclusion on this and, more importantly, how to overcome it at MMH.

While the absolute shortage of staff was indisputable, the relative shortage due to staff allocation to specific departments and work functions was not mentioned by any informant. In the outpatient department (OPD) patient flow normally ended by noon, leaving nurses unproductive. Re-thinking the organisation of care, for instance by allowing task-sharing and allocating OPD staff to work at in-patient departments in the afternoons, has the potential to alleviate some of the staff shortages. In other settings in Tanzania there has been a number of reasons for not strategically using task-sharing. These includes lack of legal or regulatory framework, but also resistance from staff to changes in job responsibilities, and lack of resources needed for coordination of a more complex way of delegating work has played a role. On the other hand, in many contexts and setting for instance in treatment of people living with HIV, task-sharing has been instrumental in scaling up treatment and making it accessible, and is widely used throughout Africa [38].

There are other factors which might affect willingness or ability to implement changes. Informants described feeling overburdened and exhausted from the chronically under-resourced work-environment where patient volume and urgent needs were high. This is known to predispose to compassion fatigue with demotivation, decreased empathy, and hopelessness [39] which in turn affects the ability to make and sustain changes.

Updated knowledge, skills and decision support were not mentioned as a significant barrier to care in our study. In other SSA countries limited knowledge and skills have been identified as barriers to acute stroke care and hypertension control [40,41] and future studies should, together with benchmarking quality of stroke care in hospitals, explore the training and guideline development needs for Zanzibar.

Belief and practices around supernatural forces and witchcraft play a big role in contemporary Zanzibar, periodically causing collective panic due to spiritual assaults [42] and spirit possessions, but is also related to everyday life and illness [43]. This was reflected in different social representations of stroke mainly as being a 'natural' versus a 'social' condition, something also found elsewhere in SSA [44] including Tanzania [45].

Explanatory models of disease is a concept that Kleinman defined to illuminate how every culture has its own set of beliefs and practices around health and ill-health. These beliefs influence how people understand and respond to illness [35]. Rather than sticking to one explanation, people draw from different beliefs and practices to make sense of their bodily experience and search for relief and healing [35].

In our study, biomedicine played a role, but often not an exclusive role for patients and their families in interpreting and acting when a stroke happened, and this has also been found elsewhere in Tanzania [45,46]. At times the biomedical explanation of stroke and the possibility of permanent disabilities clashed with the popular perceptions of stroke where stroke was seen as a spiritual malady and where supernatural treatments could cure the conditions.

The informants in our study described how they were regularly approached by patients or their families to help them determine if the experienced stroke was 'natural' or not. A 'natural' stroke referred to having a biological aetiology and should therefore be treated in hospital. On the contrary, if the stroke was not 'natural' it was thought to be a result of spirits, potentially caused by curses due to a social conflict or jealousy, and healing should be sought through traditional or spiritual practices.

It is possible that people also consulted medical doctors for treatment of the physical complications, while other types of healers were consulted for alleviation of the social and spiritual

afflictions, similar to what was found elsewhere in SSA [44,47,48]. However, it is also possible that patients and their families did their own 'trial-and-error' and concluded, if not getting a quick relief from the treatment provided in hospital, that it was not a 'natural' stroke.

Much literature in the field of medical anthropology, including by Kleinman, has illuminated the importance to engage with, rather than dismiss, local explanations and interpretations of illness and disease in order to build a patient-healer alliance based on mutual trust and respect [48–50]. Our informants, who were healthcare providers at the referral hospital, were regularly consulted to help determining if a stroke was 'natural' or not. Thereby they had the opportunity to engage with people's beliefs around stroke and appropriate stroke care. With that perspective, the practice of restricting admitted patients from pursuing spiritual, traditional, or alternative treatments may have had a valid clinical reasoning, but affected the acceptability of hospital services to the patients and their families.. A similar study of healthcare providers perception of barriers to stroke care has been conducted in Ghana [16]. The study revealed that sociocultural interpretations of stroke could influence patients to seek discharge from the hospital prematurely or avoid hospital care altogether, opting for alternative treatments. Moreover, the study noted conflicts between healthcare providers and family caregivers regarding treatment options and discharge timing.

Notable, the coexistence of different types of healthcare systems and patients engaging with more than one system implies a practice of medical syncretism. This suggests that patients, both in Ghana and Zanzibar, don't perceive these systems as distinct medical subsystems but pursue treatment as it makes sense to the individual. Additionally, healers outside the biomedical system of care might share these views and practices of their patients, and thereby be more acceptable in the view of their patients [51,52].

The healthcare access framework by Levesque is useful to structure thinking around barriers to stroke care as it includes both the health system, the population, and the interface between them. In our study we found that second to resources constraint and organization of services in the healthcare system, most identified barriers in our study were concerned with the populations ability to perceive, seek, reach, pay and engage with the health system. Other studies from SSA, focussing on the population's perspectives, have identified other barriers. These included lack of practical, emotional and social support for stroke patients and their families, and availability of services, quality of care, and provider-patient interaction [47,52–56]. Returning to the Levesque framework it also illustrates how merely focussing on the health system's side might fail to understand the preferences and actions that result from people's beliefs, values and expectations, and how these interact with healthcare access.

The findings of our study serve as a reminder of how crucial it is to consider both the perspectives of patients and healthcare systems to address critical steps in access to stroke care.

## Implications of the findings

The present findings have several important implications. Firstly, the absolute shortage of staff, available equipment and medical supplies are key barriers and need urgent attention [16,34]. If left unaddressed, the practice of individual responsibility to improvise and provide 'fixes' when resources are not available has a high cost for patients in terms of both financial cost as well as delayed admissions, premature discharges, and incomplete diagnostic investigations and treatment. While increasing resources to the health system, respectful and patient-centred healthcare must be strengthened as well as use of locally adapted best-possible stroke care protocols, covering the care continuum. Prioritization of limited resources, including staff deployment and task-sharing across departments, could potentially alleviate some urgent shortages [57].

Secondly, merely focusing on delivery side to improve access and availability might fail to understand people's lifeworld and preferential options of care. While imperative to improve stroke services, it is not enough to ensure uptake of these services. The population's health literacy levels should be increased so that competences in seeking credible health information, and making use of it, are strengthened [2,16], and a greater integration between biomedical and other health systems may be needed. As a start, we could foster curiosity and non-judgmental attitudes towards non-hospital treatments and stroke understandings to appreciate the patients' perspectives and potentially allow non-biomedical treatments alongside hospital care when appropriate [58].

Finally, while the perspectives of healthcare providers presented in this paper are pivotal to our understanding, they should not stand alone. Perspectives of other stakeholders, including patients and their families, should complement these findings as they are likely to illuminate other important obstacles to care which must be addressed, including training and competence needs, communication, and the need for the health system to support systems of care and rehabilitation within the families and communities [59,60].

## Strengths and limitations

To our knowledge, this is the first study in East Africa to explore in-depth barriers to stroke care from the perspectives of healthcare providers who are involved with stroke care provision. A strength is that we included perspectives from diverse health professions and expertise. We decreased the limitations associated with cross-cultural research by being familiar with the cultural setting, being able to conduct interviews in the local language, Swahili, when needed, and transcribing all interviews verbatim. To further limit cross-cultural misunderstandings and misinterpretations, analysis and essential findings were discussed with several local colleagues and critically revised by the Zanzibari co-author (SSS).

However, the interviewer's positionality as a medical doctor, as well as conducting research in a cross-cultural setting, might have impacted interpretation of information about the underlying social and cultural framework or missed nuances given in the interviews. Informants might have been reluctant to express views which conflicted with the biomedical understanding around stroke and might not have felt empowered to express views and opinions contrary to what they thought the interviewer was interested in or represented [61,62]. Translating the interview guide from English to Swahili could have altered some meaning of concepts and introduce bias. Interviewers with a non-biomedical background might have yielded other results and interpreted findings differently. Despite continued recruitment of informants until code saturation was achieved, we cannot exclude that other informants might have identified other barriers or weighted the contribution of them differently.

The barriers reported are those identified by healthcare providers in one particular hospital setting. Patients, their families, and other people would likely have other experiences and perspectives on barriers to care, and future studies should focus on the perspectives of patients and family members to further shed light on the barriers to stroke care. Nonetheless, the findings were in line with prior qualitative research in SSA on barriers to stroke treatment [17,33,44,47] and, we believe, contribute to the understanding of context-specific challenges of stroke management in the region.

## Conclusion

This study of healthcare providers' perspectives on barriers to timely and adequate stroke care in Zanzibar showed that several barriers to care exist related to the health system, patient-level factors, and the cultural context. On the health system level, limited resources and insufficient,

overburdened and at times burned-out staff with no specialized stroke training, together with lack of coordination of stroke services within hospital and across the tiers of health facilities, constituted major barriers to care. Patient-related factors, such as low health literacy among patients and their families, strained family relationships, financial constraints, and a perception of stroke being a hopeless, incurable condition, added to the challenges to access care. Beyond structural and financial barriers, we identified gaps between professional and lay understandings of stroke aetiology and appropriate treatments, and mistrust between healthcare providers and patients and their families.

The popular perception of stroke as a supernatural illness may in itself represent a barrier that prevents people from seeking care in hospital. Focus only on improving physical access and availability of stroke treatment in hospital may divert attention from significant cultural factors that affect healthcare seeking behaviour. If stroke care is to be improved, people's perspectives and logics of care, including self-care and traditional treatments, must be considered. Successful delivery of evidence-based stroke care requires that healthcare providers are knowledgeable about stroke and effective treatment, resourced to undertake care, that the health system they work in is supportive, and that they demonstrate cultural competency in their interaction with patients so that people can access relevant, reliable, and respectful stroke care services.

## Supporting information

**S1 Table. Example of thematic coding framework.**
(DOCX)

**S2 Text. COREQ (Consolidated criteria for Reporting Qualitative research) Checklist.**
(DOCX)

**S1 Text. Interview guide Stroke Care Providers v1.**
(PDF)

## Acknowledgment

We are particularly grateful to all the informants for the time and information they shared, and for the support received from the former and present directors at Mnazi Mmoja Referral Hospital.

## Author contributions

**Conceptualization:** Jutta M. Adelin Jørgensen, Elias Ditlevsen, Karoline Kragelund Nielsen.

**Data curation:** Jutta M. Adelin Jørgensen.

**Formal analysis:** Jutta M. Adelin Jørgensen, Elias Ditlevsen, Karoline Kragelund Nielsen.

**Investigation:** Jutta M. Adelin Jørgensen.

**Methodology:** Jutta M. Adelin Jørgensen, Karoline Kragelund Nielsen.

**Project administration:** Jutta M. Adelin Jørgensen.

**Supervision:** Karoline Kragelund Nielsen.

**Validation:** Sanaa S. Said.

**Writing – original draft:** Jutta M. Adelin Jørgensen, Karoline Kragelund Nielsen.

**Writing – review & editing:** Elias Ditlevsen, Sanaa S. Said, Richard W. Walker, Dirk Lund Christensen, Karoline Kragelund Nielsen.

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
