## [Decision Letter · Decision Letter 0]

30 Oct 2023

PGPH-D-23-01518

Understanding Healthcare Providers’ Perspectives on Barriers to Stroke Care at a resource-limited hospital in East Africa: A qualitative study from Mnazi Mmoja Referral Hospital in Zanzibar

Dear Dr. Jorgensen,

Thank you for submitting your manuscript to PLOS Global Public Health. After careful consideration, we feel that it has merit but does not fully meet PLOS Global Public Health’s publication criteria as it currently stands. Therefore, we invite you to submit a revised version of the manuscript that addresses the points raised during the review process.

Please note that we have only been able to secure a single reviewer to assess your manuscript. We are issuing a decision on your manuscript at this point to prevent further delays in the evaluation of your manuscript. Please be aware that the editor who handles your revised manuscript might find it necessary to invite additional reviewers to assess this work once the revised manuscript is submitted. However, we will aim to proceed on the basis of this single review if possible. 

We look forward to receiving your revised manuscript.

Kind regards,

Jianhong Zhou

Staff Editor

Journal Requirements:

Additional Editor Comments (if provided):

Reviewers' comments:

Reviewer's Responses to Questions

**Comments to the Author**

1. Does this manuscript meet PLOS Global Public Health’s publication criteria ? Is the manuscript technically sound, and do the data support the conclusions? The manuscript must describe methodologically and ethically rigorous research with conclusions that are appropriately drawn based on the data presented.

Reviewer #1: Yes

2. Has the statistical analysis been performed appropriately and rigorously?

Reviewer #1: N/A

3. Have the authors made all data underlying the findings in their manuscript fully available (please refer to the Data Availability Statement at the start of the manuscript PDF file)?

Reviewer #1: No

4. Is the manuscript presented in an intelligible fashion and written in standard English?

Reviewer #1: No

5. Review Comments to the Author

Reviewer #1: Overall, this is an interesting paper and commend the authors for conducting this significant research in East Africa.

Comments and suggestions for Authors:

Abstract

The abstract should be concise and well-organized, with a background, goal or objective, methodology, results, and a conclusion. The manuscript should be formatted consistently throughout.

The method section of the abstract and the manuscript must include the time and duration of the study. Also, authors should describe the theoretical approach or framework being used in the study if any.

Introduction:

The background should address the details of the previous study that had been done in the same institution and briefly describe any gaps in that research. This will eventually lead to the justification for the current study in the following paragraph, which will then lead to the primary objective of the research.

On page#3 it is indicated that that "the burden of stroke is expected to increase" (how much?). It would be more precise for readers to see the statistical data for the rate of incidence/prevalence and the mortality rate.

Before the section on the methodology, the study's objectives and purpose should be stated explicitly.

Study design:

According to the COREQ checklist, it is recommended practice to include information regarding the type of sampling in the study design.

The description of the building, the wards, and the visiting hours in the study setting section don't seem to have any relevance specific to the objective of the current study. Authors may eliminate that information and replace it with more pertinent information about the study, including the location of the data collection etc.

The characteristics and details of the study participants together with the participation selection and non-participation criteria must be included, such as years of experience working with stroke patients, etc. Also, authors may explicitly include the inclusion and exclusion criteria.

For authors reference; Tong A, Sainsbury P, Craig J. Consolidated criteria for reporting qualitative research (COREQ): a 32-item checklist for interviews and focus groups. International journal for quality in health care. 2007 Dec 1;19(6):349-57.

It is unclear whether the purposive sampling method or the snowball sampling method was the one used, therefore the information about the type of sampling that was performed must be included. Please carefully go through the information pertaining to the "informants".

Data collection section and interview guide:

Authors should provide a justification for considering conducting a semi-structured interview.

As indicated, pilot research served as the basis for the interview guide's design. It will be valuable to provide more information regarding the pilot testing for the interview guide and how the questions designed for stroke patients and their families are equally pertinent to healthcare professionals interview guides. Please refer to Page #7.

The authors should provide the interview guide as an appendix or supplementary materials.

Analysis:

The authors must clarify precisely how the data in language other than English was handled during the coding and analysis process.

Page#31: Authors should define "Biomedical ideas about stroke" and "tension between professionals"?

Discussion:

Themes and subthemes have been repeated, and authors must address new insights on the current research in more detail in the discussion section.

The arguments presented with the scientific literature are unclear and occasionally contradictory. For instance, on page#35, authors suggested task sharing as one of the potential strategies, but in the subsequent phrase, disregard the strategy by citing data from other Tanzanian settings which demonstrated the limitation of the task sharing approach in such contexts.

Second paragraph on page#37 The first line is extremely lengthy, incoherent, and lacks a clear meaning. What precisely is biomedical treatment? Does it imply receiving medical treatment?

Strengths and limitations:

Page#39: Authors must mention the details of the colleagues with whom the analysis and essential findings to limit cross cultural misinterpretations was discussed along with why and how the authors did it?

While the study appears to be sound. The language is occasionally unclear, and sentences are too lengthy, and need to be rephrased to make them more concise and simpler to comprehend. I advise the authors work with the writing coach or editor to improve the flow and readability of the text.

6. PLOS authors have the option to publish the peer review history of their article (what does this mean? ). If published, this will include your full peer review and any attached files.

**Do you want your identity to be public for this peer review?** For information about this choice, including consent withdrawal, please see our Privacy Policy .

Reviewer #1: **Yes: ** Dimple Dawar

---

## [Decision Letter · Decision Letter 1]

9 Jul 2024

PGPH-D-23-01518R1

Understanding Healthcare Providers’ Perspectives on Barriers to Accessing Stroke Care at a resource-limited hospital in East Africa: A qualitative study from Mnazi Mmoja Referral Hospital in Zanzibar

Dear Dr. Jorgensen,

Thank you for submitting your manuscript to PLOS Global Public Health. After careful consideration, we feel that it has merit but does not fully meet PLOS Global Public Health’s publication criteria as it currently stands. Therefore, we invite you to submit a revised version of the manuscript that addresses the points raised during the review process.

We look forward to receiving your revised manuscript.

Kind regards,

Bey-Marrie Schmidt, PhD

Academic Editor

Reviewers' comments:

Reviewer's Responses to Questions

**Comments to the Author**

1. If the authors have adequately addressed your comments raised in a previous round of review and you feel that this manuscript is now acceptable for publication, you may indicate that here to bypass the “Comments to the Author” section, enter your conflict of interest statement in the “Confidential to Editor” section, and submit your "Accept" recommendation.

Reviewer #1: All comments have been addressed

2. Does this manuscript meet PLOS Global Public Health’s publication criteria ? Is the manuscript technically sound, and do the data support the conclusions? The manuscript must describe methodologically and ethically rigorous research with conclusions that are appropriately drawn based on the data presented.

Reviewer #1: Yes

3. Has the statistical analysis been performed appropriately and rigorously?

Reviewer #1: N/A

4. Have the authors made all data underlying the findings in their manuscript fully available (please refer to the Data Availability Statement at the start of the manuscript PDF file)?

Reviewer #1: Yes

5. Is the manuscript presented in an intelligible fashion and written in standard English?

Reviewer #1: Yes

6. Review Comments to the Author

Reviewer #1: (No Response)

7. PLOS authors have the option to publish the peer review history of their article (what does this mean? ). If published, this will include your full peer review and any attached files.

**Do you want your identity to be public for this peer review?** For information about this choice, including consent withdrawal, please see our Privacy Policy .

Reviewer #1: No

---

## [Decision Letter · Decision Letter 2]

29 Nov 2024

PGPH-D-23-01518R2

Understanding Healthcare Providers’ Perspectives on Barriers to Accessing Stroke Care at a resource-limited hospital in East Africa: A qualitative study from Mnazi Mmoja Referral Hospital in Zanzibar

Dear Dr. Jorgensen,

Thank you for submitting your manuscript to PLOS Global Public Health. After careful consideration, we feel that it has merit but does not fully meet PLOS Global Public Health’s publication criteria as it currently stands. Therefore, we invite you to submit a revised version of the manuscript that addresses the points raised during the review process.

We look forward to receiving your revised manuscript.

Kind regards,

Nancy Angeline Gnanaselvam

Academic Editor

Additional Editor Comments (if provided):

Kindly address the reviewers' comments. While the qualitative research is rich with information, health system recommendations in resource poor settings can be added in the discussion. Authors should submit reflexivity statement.

Reviewers' comments:

Reviewer's Responses to Questions

**Comments to the Author**

1. If the authors have adequately addressed your comments raised in a previous round of review and you feel that this manuscript is now acceptable for publication, you may indicate that here to bypass the “Comments to the Author” section, enter your conflict of interest statement in the “Confidential to Editor” section, and submit your "Accept" recommendation.

Reviewer #2: All comments have been addressed

2. Does this manuscript meet PLOS Global Public Health’s publication criteria ? Is the manuscript technically sound, and do the data support the conclusions? The manuscript must describe methodologically and ethically rigorous research with conclusions that are appropriately drawn based on the data presented.

Reviewer #2: Yes

3. Has the statistical analysis been performed appropriately and rigorously?

Reviewer #2: Yes

4. Have the authors made all data underlying the findings in their manuscript fully available (please refer to the Data Availability Statement at the start of the manuscript PDF file)?

Reviewer #2: Yes

5. Is the manuscript presented in an intelligible fashion and written in standard English?

Reviewer #2: Yes

6. Review Comments to the Author

-Explain the choice of semi-structured interviews over focus groups

-Ensure that Table 2 aligns with the thematic structure in the text for consistency.

-Include a visual representation of the patient care pathway, highlighting barriers at each stage.

-Strengthen the justification for purposive sampling

Reviewer #2: (No Response)

7. PLOS authors have the option to publish the peer review history of their article (what does this mean? ). If published, this will include your full peer review and any attached files.

**Do you want your identity to be public for this peer review?** For information about this choice, including consent withdrawal, please see our Privacy Policy .

Reviewer #2: 

---

## [Editor Report · Decision Letter 3]

16 Dec 2024

PGPH-D-23-01518R3

Understanding Healthcare Providers’ Perspectives on Barriers to Accessing Stroke Care at a resource-limited hospital in East Africa: A qualitative study from Mnazi Mmoja Referral Hospital in Zanzibar

Dear Dr. Jorgensen,

Thank you for submitting your manuscript to PLOS Global Public Health. After careful consideration, we feel that it has merit but does not fully meet PLOS Global Public Health’s publication criteria as it currently stands. Therefore, we invite you to submit a revised version of the manuscript that addresses the points raised during the review process.

We look forward to receiving your revised manuscript.

Kind regards,

Nancy Angeline Gnanaselvam

Academic Editor

Journal Requirements:

Additional Editor Comments (if provided):

Kindly review PLOS Authorship guidelines and include native resident researchers who could have contributed to data acquisition for the research.

Authors are requested to read https://www.nature.com/articles/d41586-021-01795-1 on meaningful collaborations and abide by Rule 9: Be ethical and fair about publications and authorship based on the article "Ten simple rules for Global North researchers to stop perpetuating helicopter research in the Global South" https://journals.plos.org/ploscompbiol/article?id=10.1371/journal.pcbi.1009277

A detailed reflexivity statement needs to be added in methods section.
---

## [Editor Report · Decision Letter 4]

23 Jan 2025

Understanding Healthcare Providers’ Perspectives on Barriers to Accessing Stroke Care at a resource-limited hospital in East Africa: A qualitative study from Mnazi Mmoja Referral Hospital in Zanzibar

PGPH-D-23-01518R4

Dear Dr. Jorgensen,

We are pleased to inform you that your manuscript 'Understanding Healthcare Providers’ Perspectives on Barriers to Accessing Stroke Care at a resource-limited hospital in East Africa: A qualitative study from Mnazi Mmoja Referral Hospital in Zanzibar' has been provisionally accepted for publication in PLOS Global Public Health.

Best regards,

Nancy Angeline Gnanaselvam

Academic Editor

The queries have been addressed by the authors.